# Low and High Frequency Vibration Perception Thresholds Can Improve the Diagnosis of Diabetic Neuropathy

**DOI:** 10.3390/jcm10143073

**Published:** 2021-07-12

**Authors:** Tina J. Drechsel, Renan L. Monteiro, Claudio Zippenfennig, Jane S. S. P. Ferreira, Thomas L. Milani, Isabel C. N. Sacco

**Affiliations:** 1Department of Human Locomotion, Faculty of Behavioral and Social Sciences, Institute of Human Movement Science and Health, Chemnitz University of Technology, 09107 Chemnitz, Germany; claudio.zippenfennig@hsw.tu-chemnitz.de (C.Z.); thomas.milani@hsw.tu-chemnitz.de (T.L.M.); 2Department of Physical Therapy, Speech, and Occupational Therapy, Faculdade de Medicina, Universdade de São Paulo, São Paulo 05360-160, Brazil; renanlm@usp.br (R.L.M.); janesuelen@usp.br (J.S.S.P.F.); icnsacco@usp.br (I.C.N.S.)

**Keywords:** diabetic neuropathies, vibration perception threshold, mechanoreceptors, diabetes, sensitivity

## Abstract

Recent studies demonstrate neuropathic changes with respect to vibration sensitivity for different measurement frequencies. This study investigates the relationship between vibration perception thresholds (VPTs) at low and high frequencies at two plantar locations and diabetic peripheral neuropathy (DPN) severity in diabetes mellitus (DM) subjects with DPN. We examine differences of VPTs between participants with DM, with DPN, as well as healthy controls. The influence of anthropometric, demographic parameters, and DM duration on VPTs is studied. Thirty-three healthy control group subjects (CG: 56.3 ± 9.9 years) and 33 with DM are studied. DM participants are subdivided into DM group (DM without DPN, *n* = 20, 53.3 ± 15.1 years), and DPN group (DM with DPN, *n* = 13, 61.0 ± 14.5 years). VPTs are measured at the first metatarsal head (MTH1) and heel (30 Hz, 200 Hz), using a customized vibration exciter. Spearman and Pearson correlations are used to identify relationships between VPTs and clinical parameters. ANOVAs are calculated to compare VPTs among groups. Significant correlations are observed between DPN severity (by fuzzy scores) and VPTs at both locations and frequencies (MTH1_30 Hz vs. fuzzy: r = 0.68, *p* = 0.011; Heel_30 Hz vs. fuzzy: r = 0.66, *p* = 0.014; MTH1_200 Hz vs. fuzzy: r = 0.73, *p* = 0.005; Heel_200 Hz vs. fuzzy: r = 0.60, *p* = 0.032). VPTs in CG and DM groups are significantly smaller than the DPN group, showing higher contrasts for the 30 Hz compared to the 200 Hz measurement. The correlations between fuzzy scores and VPTs confirm the relevance of using low and high frequencies to assess a comprehensive foot sensitivity status in people with DM.

## 1. Introduction

Diabetic peripheral neuropathy (DPN) progressively damages the sensorial, motor, and autonomic nervous systems [1]. Among other symptoms, it causes the loss of protective sensations [2], and increases the risk of developing foot ulcers [3]. Thus, DPN leads to direct medical costs approximately five times higher than average for individuals with normal sensory perception [4]. DPN has a non-homogeneous and insidious manifestation, resulting in difficulties identifying its onset [5]. Therefore, clinicians focus on people with a high risk of ulceration, in which DPN is well established and already considered severe. While the early detection of DPN is a key factor for better forecasting an individual’s health [6], there is still no generally accepted gold standard for its diagnosis and assessment. Improving the early diagnosis of DPN and its severity in patients is important for researchers and clinicians, alike. For researchers, it is crucial to have homogeneous groups when investigating other parameters. Early and precise diagnoses enable clinicians to choose the most appropriate care and preventive measures for each patient, and most importantly, detect early impairments.

Various practicable clinical tools (128 Hz tuning fork, Semmes-Weinstein monofilaments, biothesiometry, quantitative sensory testing, questionnaires, etc.) are available to diagnose and assess the severity of DPN [5,6,7,8]. In most cases, this involves the complementary use of various tools to determine the status of different sensory modalities. In this regard, quantification of vibration perception thresholds (VPT) is considered an important marker for the assessment of DPN and may help to determine patients at risk [9,10]. However, the methodologies and their outcomes have been shown to be inaccurate [9,11]. Besides problems of reliability and objectivity, a pronounced inter-rater heterogeneity is evident. In Addition, most well-accepted methods used to assess VPTs restrict their measurements within the frequency ranges around 100 Hz to 128 Hz, mainly assessing the sensitivity of Pacinian corpuscles (FA II). However, as demonstrated by Paré et al. [12] and Shun et al. [13], both vibration-sensitive mechanoreceptors (Meissner’s corpuscles (FA I) and FA II) and their innervating Aβ-fibers are affected by DPN, showing a progression of deterioration from distal to proximal. Therefore, the use of adequate measurement frequencies is necessary for a comprehensive consideration of VPT. While 30 Hz is needed for accurate detection of distal FA I receptors [14,15], 200 Hz in turn is used for assessment of more proximal located FA II receptors [16]. Because of the different depths of the receptors within the epidermis, detection of both frequencies may provide information on the status of any loss of sensitivity.

Therefore, this study aims to examine the role of VPTs in the early detection of DPN, using low (30 Hz) and high (200 Hz) measurement frequencies for evaluating the sensitivity of both, FAI and FA II receptors of the skin [16,17]. To make advancements in the field, it is pivotal to explore strategies that focus on improving the detection, diagnosis, and management of DPN. To this end, our study pursues three main goals: (1) To investigate the relationship between DPN severity and VPT in the two frequencies and plantar locations (first metatarsal head: MTH1, heel); (2) to examine whether VPTs generated by a vibration exciter [18] differ, regarding the measurement frequency (30 Hz, 200 Hz), foot plantar location, and the presence of DM and DPN; (3) to analyze whether VPTs are influenced by clinical parameters (anthropometric, demographic (age, body mass index, sex), and DM duration).

## 2. Materials and Methods

This cross-sectional study is based on a prospectively recruited population with DM and a healthy age-matched control group (CG). All procedures were performed in accordance with the recommendations of the Declaration of Helsinki and approved by the local ethics committee of the University of São Paulo (Protocol 1.464.870). All subjects were informed in detail about the nature of the study and gave their written informed consent prior to participation. The flowchart of the study is presented in Figure 1.

DM participants were recruited from a primary care center in the city of São Paulo. Thirty-four patients with type 1 and 2 DM and 35 healthy subjects (CG) were included after eligibility criteria were checked in Process A.

The inclusion criteria for the CG were: Age between 18 and 75, no alcohol consumption within the last 12 h before the start of measurement, no DM or other diseases affecting cognition, no ulcerations/infections of the plantar sole of the foot, good skin condition of the sole of the foot, no foot/toe amputations, ability to stand and walk without assistance, no diagnosed musculoskeletal system dysfunctions, such as rheumatoid arthritis, no diagnosed heart diseases, which do not allow lying in a straight prone position.

The inclusion criteria for DM participants were: age between 18 and 75, a maximum of one amputated toe, not being the hallux, no other diagnosed neurological impairments, due to stroke, cerebral palsy, poliomyelitis, the absence of dementia, or inability to give consistent information, major vascular complications (venous or arterial ulcers), severe retinopathy, presence of plantar ulcers at the time of evaluation, and inability to walk independently for at least 10 m or the use of an assistive device. The participants with DM were further classified as with DPN (DPN group, *n* = 13) and without DPN (DM group, *n* = 20) based on a fuzzy decision support system [19]. Anthropometric, demographic, and DM type and duration of the subjects are shown in Table 1. Regarding the great individuality of the sensory system, we removed the outlier prior to the statistical evaluation. Based on Strzalkowski et al. [20], outliers defined as three times the standard deviation were removed (*n* = 8). One healthy subject and one individual with DM were excluded, due to missing values.

### 2.1. Diagnosis and Classification of DPN Severity: Fuzzy Score

The fuzzy score is a rule-based expert system to support the classification of DPN into different levels based on the disease severity [19] (http://www.usp.br/labimph/fuzzy/ingles/index.php, accessed on 1 July 2021). All participants were assessed by a trained physical therapist, who evaluated (1) vibratory perception with a 128 Hz tuning fork applied to the dorsal aspect of the hallux [8], (2) pressure sensitivity using a 10 g Semmes-Weinstein Monofilament [21] at four plantar areas (plantar surface of the hallux, first, third and fifth metatarsal heads), and (3) DPN symptoms (the Brazilian version of the Michigan Neuropathy Screening Instrument, MNSI) in both feet [22]. The sum of all scores ranges from 0 to 13 (13 represents worse DPN). Participants underwent these three clinical assessments to classify them into different severity stages, with the resulting variables used as linguistic inputs into a fuzzy model. The model performs a combinatory analysis of the input variables using an if-then rule base, linking them with the fuzzy output sets (Mamdani inference process). Those output sets are transformed into a numerical value by the center of the area defuzzification method, resulting in a DPN-degree score. The score takes on numerical values between 0 and 10 points, with 0 points representing the absence of DPN and 10 representing the most severe DPN condition. This model has shown high sensitivity and specificity in discriminating individuals with and without DPN (ROC = 0.985) [23]. Individuals with a history of ulceration were automatically classified as severe [19].

### 2.2. Vibration Perception Thresholds

VPTs were assessed under the MTH1 and heel of one foot using a customized vibration exciter (Brüel and Kjaer Vibro GmbH; type 4180, Darmstadt, Germany). Using randomization, either the left or the right foot was chosen. The probe of the vibration exciter (diameter 7.8 mm) was positioned perpendicularly to the plantar areas supported by a swivel arm. The pressure of the probe against the skin surface was controlled, keeping it within a range of 0.7 N to 1.2 N [24]. Participants pressed a button as soon as they felt a vibration of the metal probe. Subjects wore noise-canceling headphones (QuietComfort 25, Bose GmbH, Friedrichsdorf, Germany) to eliminate environmental noises. Room temperature and the temperature of the plantar surface pre and post trials were kept within acceptable ranges (±5 °C) [25]. After a ten-minute acclimatization period, every participant went through four randomized blocks (MTH1_30 Hz and MTH1_200 Hz, heel_30 Hz, and heel_200 Hz) and an additional practice trial at the beginning of the session. Each block consisted of three VPT trials to gather enough data for each frequency and location combination. A self-written LabVIEW-program (version Labview 2015, National Instruments, Austin, TX, USA) ran a customized VPT protocol inspired by Mildren et al. [26] that applies several sinusoidal vibration bursts (two seconds duration followed by a two to seven seconds pause) per trial (Figure A1) [18]. The mean of the least-recognized and the last unperceived vibration stimulus was determined as VPT.

### 2.3. Statistical Analysis

Spearman’s rank-order correlations were used to test for relationships between fuzzy scores and VPT in the DPN group. VPTs are recorded on a ratio scale, which may lead to a heteroscedastic and non-normal distribution [27]. To correct this distribution, VPT data were transformed with the natural logarithm [27]. The mean out of three VPT measurement trials per frequency and location were used for statistical analysis. ANOVAs were used to check whether there were differences in VPTs between the DM group, DPN group, and CG for all locations and frequencies tested (MTH1 30 Hz and 200 Hz; heel 30 Hz and 200 Hz). Comparisons between the two anatomical locations within a frequency were performed using t-tests for dependent samples. In addition, VPT data were analyzed descriptively with regard to DM duration (see Discussion). General linear models (GLMs) for all three groups were used to test the relationship between VPTs (MTH1 30 Hz and 200 Hz; heel 30 Hz and 200 Hz) and the anthropometric data, with age and BMI as covariates, and sex as a fixed effect. GLMs were calculated separately for each anatomical location and frequency. Pearson correlations were calculated to test for relationships between DM duration and VPTs. Multiple pairwise comparisons were corrected using Bonferroni adjustments. The statistical significance adopted was 0.05. The complete statistical analysis was performed using R 3.6.3 (The R Foundation for Statistical Computing, Vienna, Austria) [28].

## 3. Results

We found a significant and strong relationship between the DPN severity score (fuzzy score) and VPTs for both locations and frequencies for the DPN group (Figure 2).

Whereas, the CG and DM groups showed comparable VPT values, the comparison between the CG and DPN groups, as well as the DM group and DPN group, resulted in statistically significant differences (Table 2). We found significantly lower VPTs for 200 Hz compared to 30 Hz (all *p* < 0.001) at both anatomical locations and for all groups. Furthermore, we found significant differences between the two anatomical locations only for 30 Hz, for CG (*p* = 0.001), and for the DM group (*p* < 0.001) (Figure A2).

The GLMs for the CG were significant (all *p*-values < 0.001) with a residual standard error ranging between 0.64–1.32 and adjusted R-squared ranges between 0.46–0.58. The linear models for the CG explained more than 50% of the variance of the VPTs. Based on standardized coefficients, age had the greatest influence, followed by sex and BMI. Age was significant in all linear models (all *p*-values < 0.001), but sex was only significant in the heel 30 Hz condition. BMI was not significant within the models.

The GLMs for the DM group were not significant (only up to 17% of the variance of the VPTs can be explained by the model). Nevertheless, age was significant for the 200 Hz condition in both locations (MTH1 *p* = 0.032 and heel *p* = 0.034). The influence of the variables changed as follows compared with CG: age, followed by BMI, and sex.

For the DPN group, the GLMs were only significant for the 30 Hz condition at MTH1 (*p* = 0.036, up to 46% of the variance of the VPTs can be explained by the model). BMI was the only variable that was significant for this model (*p* = 0.031). All other models and variables were not significant. Furthermore, the order of influence changed as follows: BMI, followed by sex, and age. Detailed information on all GLMs can be found in Table A1.

Pearson correlation between VPTs and DM duration shows moderate and statistically significant results (Figure 3).

## 4. Discussion

Our study investigated relationships between VPTs at low and high frequencies and DPN severity in subjects with DM to identify potential advantages of its usage over the established tuning fork diagnostics. We also examined the characteristics of VPTs between participants with DM, with and without DPN, and healthy subjects. In addition, we considered differences between the whole DM group and the healthy group as a function of DM duration and associations between VPTs and anthropometric and demographic characteristics.

The main results revealed that there were high and significant correlations between the fuzzy score and the VPTs at both locations and frequencies (r ranges from 0.60–0.73) (Figure 2).

Descriptively, the DPN group showed ~5.4 times higher mean VPTs at the heel and ~9.5 times higher mean VPTs at the MTH1 at 30 Hz, and ~4.8 times higher mean VPTs at the heel and ~6.3 times higher mean VPTs at the MTH1 at 200 Hz, compared to CG and DM group.

We also found moderate and significant correlations between VPTs and DM duration (r ranges from 0.37–0.43) and descriptively higher VPTs in short time DM at the low measurement frequency. Therefore, DM seems to affect distally located neuronal structures, like FA I receptors, first. Based on our analyses, only age appears to be a variable influencing VPTs within the CG. Age, sex, and BMI cannot adequately explain the variance of VPTs within the DM group and DPN group.

In general, our results prove VPTs as elementary markers to estimate the presence of DPN [9]. In addition, vibration sensitivity should be recorded using the two receptor-appropriate measurement frequencies of 30 Hz and 200 Hz. In this context, the metric measurement results of the vibration generator provide information on even the smallest sensory changes. In combination with methods for testing other sensory modalities, DPN can thus be detected earlier.

The fuzzy score respects the minimal criteria for DPN diagnosis established by the Diabetes Study group [29], which are the presence of symptoms or signs of DPN. These include reduced plantar sensation and positive DPN sensory symptoms. (e.g., numbness while sleeping, prickling, or stabbing, burning or aching pain), predominantly at the toes, feet, or legs; or symmetrical reduced distal sensation or unequivocally reduced or absent ankle reflexes. The fuzzy score additionally includes the objective assessment of vibratory perception (128 Hz tuning fork) and tactile perception at four plantar areas (10 g Semmes-Weinstein Monofilament) in both feet.

Nevertheless, the clinically established vibration assessment methods (biothesiometry and tuning fork 128 Hz) focus solely on VPTs in higher frequency ranges (100–128 Hz), which assess mainly FA II receptors. Those current methods have also been criticized [30,31], and have been shown not to be as accurate [9,11]. Recent studies with DM participants showed VPT changes in FA I receptors stimulated by lower frequency ranges [30,31]. Thus, the use of only high frequency methods would underestimate the vibration perception alteration in this population. Therefore, better methods to assess VPTs are needed to improve the accuracy and sensitivity of the diagnosis.

Our results show high and significant relationships between the fuzzy score and VPTs measured in the DPN group. Thus, our methods could potentially improve the detection of DPN in DM participants. The measurement methodology, in contrast to the commonly chosen clinical measurements, (i) considers different locations (MTH1 and heel), (ii) captures, both types of receptors (FA I and FA II) using different frequencies, and (iii) uses a standardized contact force between the device and the measurement point.

DPN has already been related to higher VPT values [13,31,32]. Both FA I (30 Hz) and FA II (200 Hz) receptors and their pathways (Aβ) have been shown to be affected by this chronic complication [12,13]. Descriptively, our results show higher 30 Hz VPTs for both the DM group and DPN group compared to the CG. In addition, the 200 Hz VPTs of the CG were higher than those of the DM group. Nevertheless, significant VPT differences between groups were only found for CG compared to the DPN group and for the DM group compared to the DPN group (Table 2). Nevertheless, these descriptive observations appear plausible from a pathophysiological point of view. As former research has already proven [33,34], distally located epidermal structures break down first, followed by deeper dermal fibers. Furthermore, C-fibers have been found to be initially damaged by DPN [35]. Paré et al. [36] investigated the structure of FA I receptors and detected additional innervations by C-fibers in some FA I. Furthermore, they found evidence of morphological and quantitative modifications in the mechanoreceptors of the skin, which could lead to insufficient functions [12]. Their finding might explain the high VPTs in our 30 Hz data, especially for the DPN group. As DPN progresses, VPTs at both frequencies increase. Nelander et al. [30] also observed that subjects with DM are significantly less sensitive only at low measurement frequencies (8 Hz, 16 Hz, 32 Hz, and 64 Hz). To specify the foot sensitivity status more precisely, they therefore emphasize the importance of using low frequency assessments in standard diagnostics [30].

However, Lindholm et al. [31] found opposite results. The authors measured VPTs of 535 patients with type 1 DM and 717 healthy controls. Their data show significantly higher VPTs at 125 Hz in a group of participants with relatively recent DM (≤10 years). Differences at low frequencies were found in patients with longer duration DM (between 11 years and 34 years). Therefore, these frequencies can be associated with an increased the risk of developing diabetic foot ulcers [31]. Furthermore, these results may be caused by the characteristics of DM type [31]. Contrary to our study, Lindholm et al. [31] recorded only data of type 1 DM. Nevertheless, these authors point out using the combination of low and high measurement frequencies (4 Hz, 8 Hz, 64 Hz, 125 Hz) for diagnostic procedures in this population.

As described by Strzalkowski et al. [37], innervation density varies between foot regions. Therefore, we measured VPTs at the MTH1 and the heel. Our results show that the CG and DM groups were significantly less sensitive at the heel at 30 Hz, than at the MTH1. The greater presence of FA I receptors at the metatarsal area compared to the heel [37] may explain this result. For 200 Hz, we found only small differences in the VPTs between MTH1 and heel. Fairly equal densities for FA II receptors between MTH1 and the heel may be the reason for these differences [37]. In contrast to the CG and DM group, VPTs in the DPN group did not differ between MTH1 and the heel for either frequency. Structural changes in FA I and FA II receptors may be responsible for this result. Paré et al. [12] argued that structural changes occur according to DM duration and progress from distal to proximal structures [33,34]. In accordance with the literature, we found significantly lower VPTs for 200 Hz compared to 30 Hz (all *p* < 0.001) at both anatomical locations and for all groups [18,38].

Interestingly, we found moderate and significant correlations between VPTs and DM duration. To analyze the influence of DM duration in the VPT data in more detail, we categorized the VPTs in reference to Paré et al. [12]. These authors investigated processes of changing skin innervation, due to age and/or DM duration in rhesus monkeys (*n* = 12) using skin biopsies, which are an important tool for diagnosing small fiber neuropathy [39,40,41]. The authors first found hypertrophy, followed by atrophy of FA I and SA I receptors, and general structural changes of FA II corpuscles in the palmar skin with DM duration. Thus, we grouped our VPT data according to DM duration in category A: ≤4 years, category B: >4 years ≤8 years, and category C: >8 years [12]. Similar to Paré et al. [12], we observed lower VPTs at 200 Hz than at 30 Hz in subjects with DM (category A: ≤4 years) when comparing them to CG (Table 3). From category B onwards, VPTs for both frequencies were higher than those in CG.

Most likely, skin and tissue hardening, observed in participants with DM [42] spread vibration stimuli and lead to spatial summation effects [43] as a compensatory effect [44]. In general, the longer the DM duration (>4 years), the more VPTs increased for both frequencies. Nevertheless, these results should be interpreted cautiously, because the subgroups investigated in this analysis were small. Furthermore, we used the categorization based on data from rhesus monkeys, which may or may not apply humans [12].

There are many influencing factors on VPT, like the contact force of the measurement device against the skin surface [24,45] or individual characteristics of the subjects. Therefore, we decided to analyze the impact of different anthropometric parameters (age, BMI, and sex) to explain the possible differences of VPTs. We calculated GLMs divided by groups (CG, DM group, and DPN group). We only detected statistically significant impacts of age on VPTs (*p* < 0.01) for CG. As already proven, the factor age results in decreased foot sensitivity [18,46] caused by neurodegenerative processes [47]. For DM, age becomes significant for both locations at 200 Hz. However, the calculated GLM did not become significant; thus, age seems to play only a marginal role in explaining the variability of VPTs in the DM group. Interestingly, age was not an influencing factor for VPT in the DPN group, which is different than what Holowka et al. [18] observed. Furthermore, sex was the second influencing factor on VPTs in the CG.

As shown by Bergenheim et al. [47], VPT decreases earlier, and, to a greater extent, for male subjects. Again, we did not find any impact of sex on the variability of VPTs for the DM group and DPN group. The final variable, BMI, was significant only for the 30 Hz condition at MTH1 in the DPN group. As shown by [48], high BMIs affect the risk of developing diabetic foot ulcers, especially when DPN has already been diagnosed. However, the GLMs only became significant for the 30 Hz condition at MTH1. Thus, BMI seems to play only a small role in explaining the variability of VPTs in the DPN group. As a result, further factors seem to change VPTs of the DPN group related to disease progression, which we did not assess in this study and may be of interest in further studies.

However, the present study is limited to one sensory modality. Future investigations should, therefore, include other sensory modalities, such as temperature perception or pain, in addition to the measurement of VPTs with low and high measurement frequencies. Furthermore, our analysis is based on small sample sizes, especially for descriptive analyses of VPTs categorized in accordance to Paré et al. [12]. Thus, the ability to use inferential statistics to analyze the data is limited. The results of the inferential statistics performed should also be confirmed in future studies with larger and equally distributed groups. The small samples in the DM and DPN groups limit the power of the analyses, so potentially existing effects may have been masked. Furthermore, we focused mainly on patients with type 2 DM. Further investigations should involve different types of DM to compare data regarding DM characteristics. As shown in a recent comprehensive review by Viseux et al. [49], type 1 DM initially seems to present decreased functions of large Aβ-fibers. Type 2 DM initially shows C-fiber damages. Our hypothesis is that FA I break down first in type 2 DM patients, followed by FA II. This is based on the innervation of some FA I by C-fibers. However, we did not evaluate C-fiber performance directly.

The method used in this study, correlates with the DPN severity classified by a fuzzy system that combines clinical diagnostic tools (128 Hz tuning fork, Semmes-Weinstein-Monofilaments, MNSI). Our results prove VPTs are elementary markers to estimate the presence of DPN [9]. Integration of the presented measurement methodology instead of the 128 Hz tuning fork into established diagnostic procedures could improve the early detection of DPN.

## Figures and Tables

**Figure 1 jcm-10-03073-f001:**
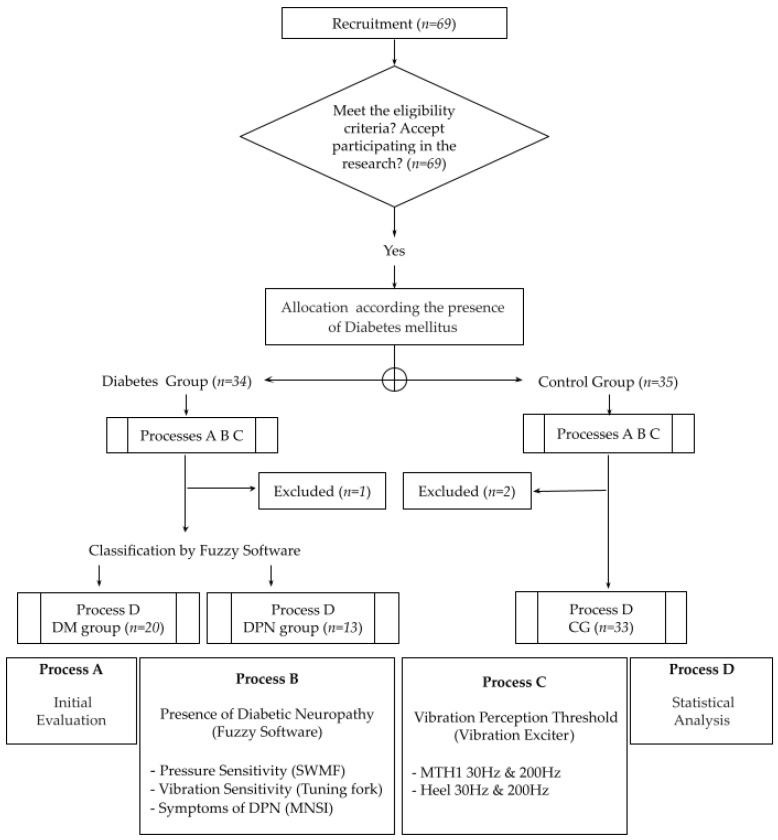
The measurement procedures included three processes: (A) Initial evaluation to check the eligibility criteria, (B) detection and classification of DPN, and (C) evaluation of vibration sensitivity. Statistical evaluation was performed as process D. SWMF, Semmes Weinstein Monofilaments; MNSI, Michigan Neuropathy Screening Instrument; VPT, Vibration Perception Threshold; DM group, participants with diabetes; DPN group, participants with diabetes and diabetic neuropathy; CG, control group; MTH1, first metatarsal head.

**Figure 2 jcm-10-03073-f002:**
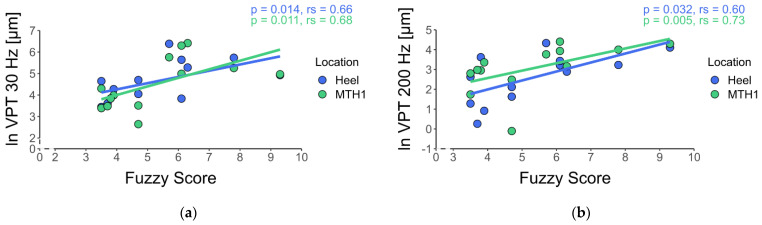
Correlation analysis between VPTs of the DPN group and fuzzy score. (**a**) correlation analysis between VPT at 30 Hz and the fuzzy score, (**b**) correlation analysis between VPT at 200 Hz and fuzzy score.

**Figure 3 jcm-10-03073-f003:**
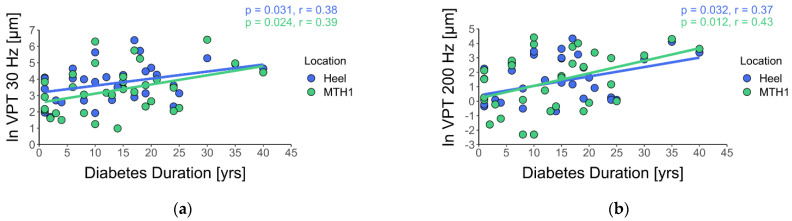
Pearson correlation between VPTs and DM duration. (**a**) correlation analysis between VPT at 30 Hz and DM duration, (**b**) correlation analysis between VPT at 200 Hz and DM duration.

**Table 1 jcm-10-03073-t001:** Anthropometric, demographic, and clinical characteristics of the control group (CG), DM group, and DPN group.

	CG (*n* = 33)	DM Group (*n* = 20)	DPN Group (*n* = 13)	*p*-Value
Age (years)	55.7 ± 15.4	53.3 ± 15.1	61.0 ± 14.5	0.241
Body height (m)	1.7 ± 0.1	1.6 ± 0.1	1.6 ± 0.1	0.049
Body mass (kg)	70.1 ± 11.9	77.9 ± 13.3	81.8 ± 17.9	0.055
Body Mass Index (kg/m^2^)	24.5 ± 3.3 *^,#^	29.5 ± 5.1 *	30.1 ± 5.8 ^#^	<0.001
Sex (female:male)	20:13	13:7	6:7	0.543
Duration of DM (years)	-	11.8 ± 10.3	17.5 ± 8.7	0.070
DM Type (Type 1:Type 2)	-	4:16	1:12	-
DPN severity (mild:moderate:severe)	-	-	7:5:1	-
Fuzzy score	-	-	5.3 ± 1.8	-
MNSI score	-	3.7 ± 2.5	4.1 ± 2.9	1.000

Data are represented as mean ± standard deviation. The *p*-value column contains *p*-values of the Kruskal-Wallis rank sum tests or ANOVAs according to the distribution and Pearson’s Chi-squared test for sex ratio. Superscripted symbols represent significant post hoc comparisons with Bonferroni adjustment: * *p* < 0.001, ^#^ *p* = 0.002. Diabetes duration was compared between the DM group and DPN group using the Wilcoxon rank sum test for independent samples.

**Table 2 jcm-10-03073-t002:** Vibration perception thresholds for the control group (CG), diabetes group (DM), and DPN group (DPN) in µm for both frequencies and locations.

	Group	VPT (µm)	*p*-Value ANOVA	Eta-Squared ANOVA	Pairwise Comparisons	
Heel 200 Hz	CG	2.0 (0.6–5.9)	*p* < 0.001	0.23	CG vs. DM	*p* = 1.0
	DM	1.1 (0.7–4.4)			DM vs. DPN	*p* < 0.001
	DPN	18.0 (5.1–31.0)			CG vs. DPN	*p* < 0.001
MTH1 200 Hz	CG	1.9 (0.4–8.6)	*p* < 0.001	0.29	CG vs. DM	*p* = 1.0
	DM	1.1 (0.5–4.7)			DM vs. DPN	*p* < 0.001
	DPN	23.7 (16.5–51.1)			CG vs. DPN	*p* < 0.001
Heel 30 Hz	CG	15.4 (7.3–31.3)	*p* < 0.001	0.37	CG vs. DM	*p* = 0.256
	DM	23.0 (14.4–57.9)			DM vs. DPN	*p* < 0.001
	DPN	104.4 (46.5–196.9)			CG vs. DPN	*p* < 0.001
MTH1 30 Hz	CG	10.7 (4.9–19.9)	*p* < 0.001	0.41	CG vs. DM	*p* = 1.0
	DM	9.9 (6.9–24.1)			DM vs. DPN	*p* < 0.001
	DPN	73.9 (33.6–191.8)			CG vs. DPN	*p* < 0.001

Sample size of the groups: CG *n* = 33, DM *n* = 20 and DPN *n* = 13. Data are represented as median and interquartile ranges. Data analysis is based on logarithmic transformation. ANOVAs with corresponding pairwise comparisons and Bonferroni adjustments were used for statistical analysis between groups. Significant differences were only found between CG and DPN, as well as the DM and DPN groups. MTH1: First metatarsal head.

**Table 3 jcm-10-03073-t003:** Categorization of VPT data (µm) by DM duration (years).

	DM Duration		*n*	Heel 200 Hz	MTH1 200 Hz	Heel 30 Hz	MTH1 30 Hz
DM and DPN groups	A	≤4	7	0.9 [0.8–2.9]	[0.6–3.0]	14.9 [10.2–43.7]	8.2 [5.9–13.7]
	B	>4 ≤ 8	4	5.4 [1.9–9.7]	6.6 [0.9–13.1]	55.9 [44.5–69.2]	27.5 [17.7–43.7]
	C	>8	22	4.7 [1.2–25.2]	12.5 [1.3–35.2]	54.2 [25.1–108.5]	35.3 [15.8–128.4]
CG		-	33	2.0 [0.6–5.9]	1.9 [0.4–8.6]	15.4 [7.3–31.3]	10.7 [4.9–19.9]

VPT data are represented as median and interquartile range and mean and standard deviation. Paré et al. [12] categories A, B, and C. Control group (CG); combined DM groups, consisting of DM and DPN groups together. MTH1, first metatarsal head.

## Data Availability

The data presented in this study are available on request from the corresponding author. The data are not publicly available due to further analyses.

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
