# Peer review of "Low and High Frequency Vibration Perception Thresholds Can Improve the Diagnosis of Diabetic Neuropathy"

_jcm, 2021, doi:10.3390/jcm10143073_

Round 1
Reviewer 1 Report
Dear author team,
Thank you for the opportunity to review your manuscript and presented data. I have read with interest your findings of low and high frequency vibration perception threshold in people with diabetes. Although there are very minor grammatical errors in the introduction and discussion sections, I have enjoyed reading your well written manuscript.
Introduction
I wonder if the introduction could further address the current screening of people with diabetes such as using a 128 Hz tuning fork together with a 10 g (Semmes-Weinstein) monofilament for the detection of diabetic neuropathy. I quite agree that these tests appear to detect diabetic neuropathy at a late, pre-ulcerative stage, requiring intensive risk management, often with poor clinical outcomes. Further, the introduction reads like it is working towards a diagnostic test accuracy paper, but the aims are focused on the severity of participants with DPN. Perhaps the advantages of using both high and low frequency testing should be emphasised in the introduction, together with the importance of this method eliminating problems such as repeatability, inter-rater heterogeneity and subjective interpretation (on the examiner’s part).
Materials and Methods
Within these participants with diabetes, how many had T1D/T2D? Also, were the biochemical markers and demographic measures used in the fuzzy score (as previously described in the referenced method) available for presentation in this manuscript? It might be helpful to the reader to present these data along with the MNSI gradings of participants in the summary section. Figure 1 is very clear and extremely useful for gaining an overview of the study protocol.
Results
Unfortunately, I was unable to login and access the fuzzy score methodology that was linked in the manuscript. I understand that it is based on the MNSIQ and MNSIE along with other anthropometric, clinical and demographic measures. Of the 13 participants with DPN, how many participants were classified as mild/moderate/severe based on the fuzzy score? The high and low frequency vibration perception thresholds are much greater in participants with DPN compared to participants with diabetes only. Given the long diabetes duration in participants with confirmed DPN, it seems important that vibration perception threshold can identify sub-clinical deficits for screening and also in those individuals who satisfy the minimum clinical diagnostic criteria of DPN with either + signs in the presence or absence of symptoms.
Table 2 is not particularly clear. Perhaps for clarity additional columns could be added to indicate the p-value and the group comparison included in the ANOVA. Moreover, the difference in vibration perception threshold appears to be so much greater in participants with DPN. Are these participants classified in the early stage of the clinical course of DPN?
Nitpicking
I appreciate it is a very small thing, but when introducing a researcher’s work it is best to indicate their name such as “Smith et al [REF]”. This might be helpful to the reader instead of having to look up the reference in the list each time.
The use of the reference Paré et al for data of intra-epidermal nerve fibre density in hominids with diabetes might better be found in the following publications:
Divisova S, Vlckova E, Srotova I, Kincova S, Skorna M, Dusek L, Dubovy P, Bednarik J. Intraepidermal nerve-fibre density as a biomarker of the course of neuropathy in patients with Type 2 diabetes mellitus. Diabet Med. 2016 May;33(5):650-4. doi: 10.1111/dme.12890. Epub 2015 Sep 7. PMID: 26344697.
Collongues N, Samama B, Schmidt-Mutter C, Chamard-Witkowski L, Debouverie M, Chanson JB, Antal MC, Benardais K, de Seze J, Velten M, Boehm N. Quantitative and qualitative normative dataset for intraepidermal nerve fibers using skin biopsy. PLoS One. 2018 Jan 25;13(1):e0191614. doi: 10.1371/journal.pone.0191614. PMID: 29370274; PMCID: PMC5784950.
Lauria G, Bakkers M, Schmitz C, Lombardi R, Penza P, Devigili G, Smith AG, Hsieh ST, Mellgren SI, Umapathi T, Ziegler D, Faber CG, Merkies IS. Intraepidermal nerve fiber density at the distal leg: a worldwide normative reference study. J Peripher Nerv Syst. 2010 Sep;15(3):202-7. doi: 10.1111/j.1529-8027.2010.00271.x. PMID: 21040142.
I am not affiliated with the suggested references in any way.
Author Response
Dear Reviewer 1,
thank you very much for your comments.
We have summarized our responses in the attached document.
Yours sincerely,
Tina J. Drechsel

Reviewer 2 Report
Thank you for the opportunity to review this paper. Please read my comments below.
Abstract
- Line 15: Please provide one or two sentences that describe the background of "why" you are measuring the relationship between low and high frequency VPT and severity of DPN.
- Line 16 and 24: Clinical parameters and DPN severity? Is it meaning the same? Please be consistent.
- Line 19: CG, please state that this is an acronym of 'control group' as it can be different in other contexts.
- Line 21: DMg and DPNg is confusing as "group" was not stated in the defined acronym. Please provide an acronym that could be universally accepted and define what it means (you can simple change "individuals with DM and DPN" as DPN group).
- Line 26: Please provide r value (with p-value) for each of the correlation, not as a range.
- Line 27: "showing higher contrasts for the 30Hz measurement" meaning compared to 200Hz? And contrasts meaning the difference between the groups? Please clarify.
Introduction
- First paragraph (line 34-47): This paragraph is very long. Please distinguish the main topic sentence (DPN problem vs need of diagnosis) and separate the paragraphs.
- Line 42: Do you not agree nerve conduction measure as a gold standard?
- Can you provide more detailed information and rationale for choosing low as 30Hz and high 200 Hz? And importance of adding other nerve unit types?
- If you want to argue that additional frequency measures could help early detection of DPN, I think you should also compare with the frequency commonly used, which is 128 Hz. I wonder the results of most acceptable methods (100Hz to 128Hz) and your measures of 30 Hz and 200 Hz are different or not? Have you sought this relationship?
- Line 55: How can adding two different frequency "early detect" DPN? Do you think your current comparisons and design of the study could detect earlier than what has been previously used? How? Does identifying different receptors could be more sensitive on detecting the nerve damage? Please provide more rationale and help readers understand what you are trying to do.
- Second paragraph (Line48-65): Can you also separate the information with the main topic sentence? Please separate out the paragraph with the study aims.
- Line 61: MTH1 is not defined (defined later, line 126).
Materials and Methods
- Line 69: CG is not defined. Should be defined here, not at line 82.
- Line 83: You recruited both DM/DPN and healthy subjects based on the inclusion criteria. I don't see the inclusion criteria for the healthy group. Please include this.
- Please provide more detailed information about the two healthy outliers. VPT low? high?
- Please perform a statistical test (chi-square) for the sex ratio.
- Line 118: fuzzy score (0-10 points). 10 indicates what? Please describe.
- Line 128: How did you randomize? And was randomization matched between DMg, DPNg and CG?
- Line 136-138: Was there a rest-period between the blocks? Was the 7 seconds pause (line 141) the rest period between the trial?
- Line 140: reference 23. Please provide the author name with the reference.
- Line 141: I don't understand what value you are trying to average.
- Line 145: After transforming VPT value to natural log, what does it mean?
- Line 146: Were the assumptions for ANOVA met?
- Line 146-153: Can you make subheading for statistical analysis and separate out this information?
- Line 148: Please provide more information about the GLM. For the VPT, were all the locations and testing frequency used?
- What software/version did you use for the statistical analysis? Please include this information.
- Line 148: Were number of sample size adequate to perform GLM for each group?
Results
- Line 150: Why did you also covary for age?
- Line 152: Assumptions for Pearson correlation was met?
- Line 155: strong relationship? reference for stating this?
- Line 155: Please be consistent on the order of the findings. I expect ANOVA results first since you stated that analysis first in the previous paragraph.
- Line 158: what is acceptable range? Please provide the detailed numbers.
- Line 158-160: Please provide more detailed results for this and revise table 2.
- Line 160-162: Please remove "in accordance to the literature" and references 14, 25 as results section should contain only the findings of your study. You can add this in your discussion.
- Line 162-164: Your statistical analysis states that you only compared between the groups. However, the results show that there was a difference between the locations. This is confusing. Please be clear on what stats you conducted and the results related to the analysis.
- Line 165: Table 2 has too many symbols which is extremely difficult to understand. The meaning of the each symbol has not been described. Please revise table 2 with more clear description. If you used ANOVA, why is the data provided with median and IQR?
- Line 172: Can you state each of the specific information rather than the range? If there is too much information , please provide as an appendix.
- Line 174: provide the exact variance rather than saying more than 50% or up to 17%.
- Line172-186: Difficult to understand the results. Please provide the information with consistency and avoid using "all p-values <0.001" as what "all" states is not clear.
- Line 187: This information should be in methods section (statistical analysis).
- Line 191-193: This information should be in methods section. Provide author name for [26] and the reference.
- Line 195: parenthesis missing at the first row of table. Table 3 also has median and IQR. If you want to state this in discussion, please provide the results in this section. This all of sudden shows up in the discussion, which is not explained well enough here.
Discussion
Throughout discussion and entire manuscript. Please provide author names with references rather than just writing the reference number.
- Line 207: reference for stating high correlation?
- Line 220: How does using both frequency detect DPN "earlier"? I think this conclusion is a stretch.
- Line 200-218: Mostly repeating results. Please provide the significance of the study rather than redundantly stating the results.
- Line 230-237: Consider including these information in the introduction. Some information that should be in introduction are missing, which would provide a good rationale for the study.
- Line 227 and Line 232: Fuzzy score states 128Hz as objective assessment, but then states it has been criticized and not accurate? This information is contradicting and confusing.
- Line 244-Line 259: Some of information here should also be in introduction to help readers why you chose the two frequencies that could help understand the significance of this study.
- Line 280-281: Method section
- Line 281-284: does not support well why you subcategorized your findings.
- Table 2 between DMg vs CG is not statistically compared. Thus, I don't find the reason for explaining why DM had lower value then CG. Is this finding added after you've sought the results?
- If CG has higher VPT in both 30Hz and 200Hz at two locations compared to DMg+DPNg, how could this method help early detection of DPN?
- Line 299: remove e.g.
- Line 309: IF high BMI is a risk for developing ulcer, why did BMI only showed as a significant factor at 30Hz at MTH1? Any additional thoughts?
- Line 319: If the VPT difference between the groups were greater in 30Hz, why do you recommend using both?
- Line 323: You stated you mainly focused on type 2 DM, but I cannot find how many people were type 1 or 2 in your study. Please provide how many individuals were type 1 and type 2 for each DM and DPN group.
- Please provide conclusion of this study after the limitations (unless journal requires limitations to be the last).
Author Response
Dear Reviewer 2,
thank you very much for your comments.
We have summarized our responses in the attached document.
Yours sincerely,
Tina J. Drechsel

Round 2
Reviewer 2 Report
Dear authors,
Thank you for the thorough response to the comments. The manuscript has been well elaborated with more clarity. I still have some comments, but it could be addressed easily. Please see my comments below.
Introduction
I really like the improvements made in the introduction.
Methods
Line 109-113 starting with “or other diagnosed… the use of an assistive device”: Are these inclusion criteria or exclusion criteria? I am slightly confused that you want to include people with stroke, CP, polio with DM? Wouldn’t that confound the interpretation of the outcome?
Please see my previous comment about line 141: From your response, I believe Figure 2 (Measurement algorithm for a measurement trial) provides useful information on understanding your methods. If this figure is made by your team, can you submit this figure (with more detailed descriptions) as an appendix? It will be helpful.
Line 174: Please remove “Statistical significance adopted was 0.05” since you state this at the very end again. Maybe you can state as “All statistical significance was adopted at 0.05.” for more clarity. And it seems you had multiple pairwise comparisons in your study. So how did you control type 1 error?
Line 196: I asked about an ‘acceptable range’ to understand what you stated in the response. I don’t think you need to provide plantar temperature degrees in the results (unless the other reviewer asked for it). This temperature comparison was not mentioned in your statistics and could be moved to the method section for more details (after line 161) or could be deleted from the manuscript.
Table 2. This table still needs more elaboration. If * ° # + § $ these symbols mean p<0.001, why do you want to use different symbols? Same for superscripted 1 and 3. It seems like your intention was that each symbol means ‘significant difference of X and Y’ of either different location comparison or different frequency comparison? You should provide clarity on what “significant difference” you are trying to explain. It is still an extremely difficult table for readers to follow. For the pairwise comparison column, I don’t think italic and bold font helps readers understand the comparisons. Please divide the columns with 3 comparisons (CG vs. DM, CG vs. DPN, DM vs. DPN) and provide the non-significant pairwise values (CG vs. DM) as well. For other comparisons, you can use appendix or
Discussion
Line 317-326: I appreciate your effort in including the author’s name. But, you are now overstating their name in this paragraph for 4 times. Please elaborate on this paragraph more thoughtfully.
Please see your previous response about line 220: “In addition, our results suggest using both frequencies to more accurately assess DPN status in terms of vibration sensitivity in individuals with DM, potentially detecting DPN earlier.” I did not find the revised sentence in your manuscript. Has it been deleted?
Minor changes recommended:
Line 23: I think it should be “DPN group (DM with DPN …”
Line 153: No need to re-define MTH1.
Line 180, 184, 257. 258: MTH1, not MTH.
Line 210: where --> "were"
Line 343: age and/or DM “duration”
Author Response
Dear Reviewer 2,
attached you can find our comments to your previous comments and suggestions.
Kind regards,
Tina J. Drechsel
